# Simplifying Prediction of Intended Grasp Type: Accelerometry Performs Comparably to Combined EMG-Accelerometry in Individuals With and Without Amputation

**DOI:** 10.3390/s25226984

**Published:** 2025-11-15

**Authors:** Samira Afshari, Rachel V. Vitali, Deema Totah

**Affiliations:** Department of Mechanical Engineering, University of Iowa, Iowa City, IA 52242, USA; rachel-vitali@uiowa.edu (R.V.V.); deema-totah@uiowa.edu (D.T.)

**Keywords:** accelerometry, electromyography, hand gesture recognition, wearable sensing, prosthetics

## Abstract

The adoption of active upper-limb prostheses with multiple degrees of freedom is largely lagging due to bulky designs and counterintuitive operation. Accurate gesture prediction with minimal sensors is key to enabling low-profile, user-friendly prosthetic devices. Wearable sensors, such as electromyography (EMG) and accelerometry (ACC) sensors, provide valuable signals for identifying patterns relating muscle activity and arm movement to specific gestures. This study investigates which sensor type (EMG or ACC) has the most valuable information to predict hand grasps and identifies the signal features contributing the most to grasp prediction performance. Using an open-source dataset, we trained two types of subject-specific classifiers (LDA & KNN) to predict 10 grasp types in 13 individuals with and 28 individuals without amputation. Having 4-fold cross-validation, LDA average accuracies using ACC only features (84.7%) were similar to combined ACC & EMG (88.3%) and much greater than with only EMG features (58.1%). Feature importance analysis showed that participants with amputation reached more than 80% accuracy using only three features, two of which were ACC-derived, while able-bodied participants required nine features, with greater reliance on EMG. These findings suggest that ACC is sufficient for robust grasp classification in individuals with amputation and can support simpler, more accessible prosthetic designs. Future work should focus on incorporating object and grip force detection alongside grasp recognition and testing model performance in real-time prosthetic control settings.

## 1. Introduction

Hand gesture recognition has been extensively used for both descriptive (nonverbal) communication and functional grasping applications [1]. Vision or image-based gesture recognition is popular for communication applications [2], including sign language recognition [3,4]. However, image-based recognition is impractical for functional grasp detection, such as in assistive robotics, due to its limitations in real-world scenarios where video or image-based sensing is not practical. Wearable sensors are a promising alternative for gesture recognition in functional applications (e.g., prosthesis control [5]), and facilitate gesture recognition outside a laboratory environment where the visual access needed for image-based recognition is often limited.

Wearable sensors integrated into assistive technologies aim for intuitive interactions and enhanced user experience, particularly for individuals with disabilities. In upper-limb prosthetic devices, however, many market-available myoelectric (i.e., muscle-activated) hands offer only one or two degrees of freedom (DOFs) like opening/closing the hand or pronating/supinating the wrist [6]. Although these functions offer beneficial yet restricted utility, multi-DOF prosthetic devices more accurately replicate natural hand movements. Wearable sensors can play a crucial role in enabling such advanced control by capturing movement and muscle activity data for complex hand gesture recognition.

Several types of wearable sensors are commonly used for hand gesture classification, which tend to focus on muscle activity and kinematics measurements. Muscle activity-based recognition uses electromyography (EMG) sensors [7,8,9,10,11,12,13,14,15,16], while limb kinematics are captured with accelerometers (ACC) [7,8,9,10] or inertial measurement units (IMUs) [11,12,17,18,19], which include triaxial ACC and angular rate gyroscopes as well as magnetometers in many cases. EMG signals are useful for distinguishing subtle gestures with fine finger movements [13]. On the other hand, classifiers based on IMUs (including ACC sensors) can identify a variety of motions ranging from command-type communicative gestures of bed-ridden patients [19] to object manipulation gestures for activities of daily living (e.g., eating, drinking, and answering phone) [17]. ACC data is less affected by amputation than EMG [7], suggesting that prosthesis performance may be improved by the combination of EMG and ACC sensors.

Multisensor approaches consistently improve hand gesture classification accuracy, particularly for able-bodied individuals [8,9,10,12,20,21]. Sensor combinations vary, with some incorporating palm pressure sensors [12] or gaze data [10]. Studies specifically focused on combining EMG and inertial data generally report improved performance, with accuracies ranging from 77% [7] to over 90% [22,23] for able-bodied participants. Interestingly, Gijsberts et al. (2014) combined EMG and ACC data for able-bodied individuals to achieve a classification accuracy of 83%, which was an improvement compared to using only EMG (77%) or ACC (81%) [8]. Some work has included participants with amputation, showing slightly lower but still promising results of 61% for 5 participants [7], 77.8% for 2 participants [21], and 89.6% for 11 participants [23].

While these results demonstrate the potential of multimodal sensing, they do not fully address the practical challenges faced by individuals with upper-limb amputation in daily prosthetic use. A notable proportion of patients choose not to use upper-limb prostheses in daily life, with studies reporting a device abandonment rate of approximately 20% due to high cost, complexity, and unreliable and/or counterintuitive multi-DOF control [24]. Most commercially available devices do not have sufficient DOFs to effectively mimic natural limb movements, and issues such as weight, discomfort, and limited functionality cause device abandonment [25]. A qualitative study found that many prosthesis users continue to rely on their sound limb, citing the unreliable nature of myoelectric prostheses, even with advanced multi-function devices [26]. While pattern recognition is often perceived as a more natural control method compared to conventional strategies, it requires extensive training and lacks robustness in daily use. Addressing key challenges associated with wearable sensors can help enhance the usability of prosthetic devices. Efficient sensor use and advanced data processing techniques are necessary for accurate and responsive control systems using wearable sensors, such as surface electromyography (sEMG) electrodes [27] and ACC sensors.

Most existing datasets primarily include able-bodied participants, offering limited insights into the performance of gesture recognition systems for individuals with amputation. Moreover, most studies evaluate combined modalities without isolating each sensor’s individual contribution to classification performance. This approach limits understanding of which sensor types provide the most valuable information for gesture recognition, and limits a detailed understanding of each sensor’s impact on performance.

Moreover, the observed performance disparities between these populations, along with variability across datasets, underscore the challenges in achieving robust models applicable to users with amputations. Notably, Krasoulis et al. (2017) is among the few exceptions, offering analyses of sensor type contributions and demonstrating that multimodal systems, especially with minimal sensor setups, can achieve reliable real-time prosthesis control, including data from both groups of subjects, those with and without amputation [21]. Building on this work, Krasoulis et al. (2019) demonstrated a real-time implementation using only two EMG-IMU sensors, incorporating a confidence-based rejection strategy to enhance reliability during multi-grip prosthesis control in both able-bodied and amputee participants [5]. However, the number of participants with amputation in those studies was limited to a maximum of two, highlighting the need for further investigation with larger and more diverse amputated populations. These limitations highlight the need to not only improve accuracy but identify the most informative signals for enhancing real-world applicability and reliability of gesture recognition systems for prosthetic control.

The overarching goal of this work is to identify the minimum sensor configuration required for accurate grasp classification by identifying the most valuable data from wearable sensors. Importantly, the current work analyzes data from able-bodied participants and ones with trans-radial amputation to address two main questions: (1) which modality (EMG or ACC) is more informative for predicting grasp types for each population of participants and (2) which specific signal features contribute most to classification accuracy for each population? Answering these questions involves identifying a minimal dataset that maximizes classification accuracy while minimizing complexity. The findings are expected to provide insights into the optimization of sensor data utilization for prosthetic control, in both able-bodied and Amputee populations. This could enable more cost-effective and user-friendly prosthetic devices design, increasing the likelihood of prosthesis adoption and effectiveness.

## 2. Method

### 2.1. Dataset

To compare gesture recognition performance of EMG and ACC sensors, we used an open-source dataset (MeganePro) [10], which includes data collected from 15 individuals with trans-radial amputation and 30 able-bodied participants. Two participants with amputation and two able-bodied participants were excluded from the analysis in this study due to missing/unreliable sensor data. The EMG and tri-axial ACC data utilized in this study were collected from 12 electrodes (Trigno Wireless sEMG system, Delsys Inc., Natick, MA, USA) that were placed in two arrays around the forearm or residual limb. One array of eight electrodes started at the radio-humeral joint, with a second array of four electrodes placed 45 mm distally aligned with the gaps of the first array, and without targeting specific muscles (Figure 1). In total, there were 12 channels for EMG and 36 channels of ACC (X-, Y-, and Z-axis) data.

The dataset was collected while participants completed 10 different grasp types (Figure 2) with 18 different objects in two conditions, *static* and *dynamic*. To collect *static* data, participants were instructed to grasp the objects without moving or lifting them. In the *dynamic* condition, participants used a grasp to perform a functional task involving an object. These movements replicated activities of daily living, combining different grasp types with objects, such as holding a can with a medium wrap while drinking. Participants performed tasks in seated and standing positions for *static* trials (4 repetitions of seated, 4 repetitions of standing, for 8 repetitions overall) and either seated or standing position (4 repetitions) for *dynamic* trials. The choice of either a seated or standing position was determined by how the action is typically performed in real life. For example, a door is often opened while standing and a button on a remote is often pressed while seated [10].

### 2.2. Feature Extraction, Feature Evaluation and Classification

Prior to feature extraction, for each sensing modality, both EMG and ACC signals were divided into time windows, each including 400 data samples and with 95% overlap between successive windows [10]. Next, a comprehensive set of features commonly used in hand gesture recognition (Table 1) was calculated for each time window. Features were extracted without normalization to avoid participant-specific scaling that would require individual calibration in real-world prosthetic applications.

All features listed in Table 1, for both EMG and ACC in each axis, were scored using feature importance metrics based on participant-specific decision trees. The MATLAB (MathWorks, R2023a, Natick, MA, USA) function “fitctree” was used to form a decision tree, and the function “predictorImportance” produced estimates of the importance score for each feature. Each feature score was then divided by the summation of all feature scores and multiplied by 100, thereby reported as a percentage among all feature scores. Extracting features for each electrode and axis resulted in 468 total predictors: 144 EMG features (12 electrodes × 12 EMG features) + 324 ACC features (12 electrodes × 3 axes × 9 ACC features) to train the machine learning model. Features were then ranked from the most important feature to the least, based on the mean importance score percentage across participants within each participant group (with or without amputation).

Linear Discriminant Analysis (LDA) classification model is mainly used in this work, since it is one of the most common classifiers in gesture recognition and has been shown to achieve high classification accuracies (>65%) [7,10,13,15,28,29]. Subject-specific LDA models were trained and tested using the top feature as an input, then retrained after adding the next best feature and so on, until classification accuracy for the entire feature set was tested. This process was repeated using 4-fold cross-validation. In each fold, two of the eight *static* repetitions and one of the four *dynamic* repetitions were separated into the test set. LDA models were trained with the remaining repetitions. Prior to model training, hyperparameter tuning was performed using nested 3-fold cross-validation within the training set, where repetitions from each participant were split into folds to avoid overlap with test data. However, the optimized hyperparameters were found to match MATLAB’s default settings, so the defaults were retained for model training.

To assess the robustness of results against the chosen classifier, a KNN (K = 5) classifier was also trained on this dataset, with all EMG and ACC features in Table 1. Data split for train and test KNN model was same as the approach described above for LDA, based on the number of repetitions.

All 468 predictors received importance scores as percentages, and averaged across individuals. These scores were then averaged through all channels and sorted from the highest to lowest average percentage score through all channels. During this whole process, feature importance scores were sorted within the two groups of participants (with and without amputation), and in three cases according to the input sensor (EMG only, ACC only, both EMG and ACC). The sorted order of feature scores also determined the order of adding features to the classifier to obtain accuracy boxplots. In other words, accuracy boxplots across participants for each modality were obtained adding features in the descending order of sorted mean feature importance percentage. The classification accuracy was calculated by dividing the number of correctly classified grasps by the total number of predictions in the test set [30]. In addition to overall accuracy, percentage recall values for each grasp type, as well as confusion matrices for the LDA model were generated to visualize class-specific performance and identify common misclassifications across grasp types.

After obtaining the accuracy boxplots with adding features one by one, the most important features for each data type were selected based on the difference in median accuracy across participants. The median accuracy difference for each additional feature was compared to determine an appropriate stopping point for selected features, where the accuracy increase plateaued. The selection process continued until the median improvement in classification accuracy between successive feature additions fell below 1 percentage point, which served as the numerical stopping threshold for identifying the most informative feature subset.

All computations were performed on a desktop computer equipped with an Intel Core i9-12900K CPU, 128 GB RAM (Intel Corp., Santa Clara, CA, USA) and running Windows 11 (Microsoft Corp., Redmond, WA, USA). Model training time for each modality and subject group is calculated and reported to capture the time spent training the LDA model, excluding the time required for feature computation, feature evaluation, data loading, and hyperparameter optimization.

### 2.3. Statistical Analysis

Two-way mixed-design ANOVA was implemented for LDA accuracy values using all features. ANOVA was conducted using accuracies obtained from the four-fold cross-validation procedure, treating each fold as a repeated observation. The between-subject factor (Participant Group) distinguished between able-bodied and amputated participants, while the within-subject factor (Sensor Modality) represented the three conditions: EMG, ACC, and EMG+ACC. Mauchly’s test of sphericity was performed to evaluate the sphericity assumption; when violated (*p* < 0.05), Greenhouse–Geisser-corrected *p*-values were reported. Bonferroni-corrected post hoc tests were subsequently applied to identify pairwise differences between modalities. This analysis allowed quantification of both the main effects (Participant Group and Sensor Modality) and their interaction (Group × Modality) on classification performance, thereby determining whether performance differences across sensing modalities and participant groups were statistically significant (α = 0.05). Effect sizes were quantified using partial $\eta^{2}$ for all ANOVA effects.

## 3. Results

ACC alone and combined EMG+ACC both outperformed EMG alone, for both participant groups (Figure 3). Specific LDA classification accuracy values varied by sensor modality and participant group. Using EMG alone achieved 62.0% mean accuracy for able-bodied participants and 54.1% for those with amputation. ACC alone achieved 85.0% and 84.4% respectively, and using EMG+ACC resulted in mean accuracies of 89.1% and 87.5% respectively. Results obtained with the KNN model showed a similar pattern, with the ACC and EMG+ACC modalities outperforming EMG alone. Across all subjects (both groups), the average KNN accuracy was 47.7% when trained with EMG only, 68.7% when trained with ACC only, and 80.6% when trained with both EMG and ACC combined (Figure 3). From this point onward, the reported classification accuracies refer to those obtained with the LDA classifier, given its consistently better performance compared to KNN. In other words, unless otherwise specified, all accuracy values discussed in this work correspond to the LDA results. Detailed per-subject accuracies for both groups are presented in Appendix A, highlighting inter-individual variability.

LDA confusion matrices (see Appendix A) show that misclassified grasps were generally distributed across several other labels, rather than concentrated on a single incorrect class. However, certain grasp pairs were consistently confused, such as prismatic pinch vs. precision disk and lateral vs. tripod, which share similar thumb–finger coordination and motion patterns. These overlaps likely drove much of the accuracy loss across modalities.

Using selected features reduced accuracy minimally: EMG (56.6% vs. 51.1%), ACC (84.5% vs. 83.9%), and EMG+ACC (85.7% vs. 82.7%) for able-bodied vs. amputation groups. Classification accuracy tends to plateau after adding 5 features for EMG, 4–5 features for ACC, and 3–9 features for combined modalities (Figure 4). Table 2 shows the selected set of features for each sensor and participant group. Features were selected in the order of their average feature importance score (Figure 5) across sensor channels (see Appendix A for importance scores for each channel).

EMG mean absolute value (MAV) was the most important feature, whether considering EMG data alone or in combination with ACC data, regardless of participant group. Also, EMG waveform length (WL) is the second most important feature for EMG data in both groups of participants, and only able-bodied participants, when combined with ACC data. The list of selected features for EMG data is the same for both groups of participants, with a slight difference in the order of RMS and VAR features (Figure 4a). For ACC data, selected features are similar across both participant groups, but the order of importance differs. MAV, in particular, is selected only for able-bodied participants, because of the difference adding this feature made in classification accuracy (Figure 4b). When EMG and ACC features are combined, only three features overall are enough to obtain high enough classification accuracy in participants with amputation. In this group, the first feature is MAV from EMG data, while the next two top features are ACC ones. For the able-bodied group, nine features were selected, including both EMG and ACC ones (Figure 4c).

Training time analysis shows that using a selected subset of features substantially reduced the classifier’s training duration in both participant groups, while still maintaining high classification accuracy (Table 3). The largest reduction was observed in the EMG+ACC condition for able-bodied participants, where training time dropped from 17.91 s to 5.16 s.

### Statistical Analysis Results

The mixed-design ANOVA revealed significant main effects of Participant Group (*p* < 0.001) and Sensor Modality (*p* < 0.001, Greenhouse–Geisser-corrected), as well as a significant Participant Group × Sensor Modality interaction (*p* < 0.001). The main effect size of Sensor Modality was very large, accounting for 96.5% of explained variance (partial $\eta^{2}$ = 0.965), while Participant Group and Group X Modality interaction accounted for 11.5% (partial $\eta^{2}$ = 0.115) and 27.6% (partial $\eta^{2}$ = 0.276) of the variance, respectively. Post hoc comparisons (Bonferroni-corrected) indicated that EMG+ACC achieved significantly higher accuracies than either EMG or ACC alone (*p* < 0.001), and ACC significantly outperformed EMG alone (*p* < 0.001).

## 4. Discussion

Overall, combining accelerometer and EMG modalities had the highest accuracy for participants with or without amputation. The KNN results show a similar overall trend to those with the LDA model (Figure 3), indicating that the relative informativeness of each sensing modality is robust across classifiers. However, EMG contributes more to the KNN model, as evidenced by the performance difference between ACC and EMG+ACC, which was not observed with LDA. In other words, KNN required both EMG+ACC inputs to achieve performance comparable to that of LDA using ACC alone. Accelerometry (ACC) proved highly informative for grasp classification, outperforming EMG alone and closely matching the performance of combined EMG+ACC models, when using an LDA classifier. These results align with prior offline studies demonstrating the added value of ACC in improving classification accuracy for able-bodied participants [8,22], and with real-time findings demonstrating the effectiveness of multimodal control using minimal sensor setups [5,21]. It should be noted that the studies referenced for comparison employed varying sensor configurations, electrode placements, and experimental tasks. For example, Wang et al. (2023), Gijsberts et al. (2014) and Atzori et al. (2014) used the publicly available dataset NinaPro [7,8,22], while others, Krasoulis et al. (2017), Krasoulis et al. (2019), Khomami et al. (2021), and Tripathi et al. (2023), collected their own data with different measurement setups to train and test classifiers [5,11,16,21]. Nonetheless, the majority of their participants were able-bodied.

Importantly, our findings reveal that the contribution of ACC is even more pronounced for individuals with upper-limb amputations, who exhibited lower EMG-only accuracy than their able-bodied counterparts. While our ACC (85%) and EMG+ACC (89%) accuracies surpassed those reported by Gijsberts et al. (2014) (ACC: 81%; EMG+ACC: 83%), our EMG-only accuracy for able-bodied participants (62%) was notably lower than theirs (78%) [8]. This discrepancy may stem from differences in participant populations, machine learning models, and task complexity. For example, Gijsberts et al. (2014) used Kernel Regularized Least Squares (KRLS) to classify offline 40 distinct hand and wrist movements in NinaPro DB2 dataset consisting only of able-bodied participants [8]. Our EMG+ACC classification accuracies also surpassed EMG+IMU accuracies reported by Krasoulis et al. (2017) [21]. Their real-time median classification accuracies were 82.7% for able-bodied and 77.8% for amputee participants using an LDA classifier on data from 20 able-bodied and only 2 amputee participants [21]. Their dataset included data collected from an armband during the experiment while subjects performed 12 gestures with no object interaction. Differences in measurement setups and data acquisition protocols may partly account for the observed variation in reported accuracy values across studies.

Our findings are also aligned with prior research on sign language gesture recognition, which demonstrated that ACC data outperformed EMG for classification tasks [11]. Their study involved 10 able-bodied participants performing 20 sign language gestures and evaluated both EMG and IMU signals. In that study, sensors were placed on 4 targeted muscle spots on the forearm. Feature selection revealed that the majority of informative features came from the IMU, highlighting its strong contribution to classification performance. However, their analysis did not isolate the contribution of ACC alone or include participants with amputation. Overall, these results underscore the general utility of ACC signals in hand gesture recognition.

While prior work has highlighted the benefits of combining EMG and ACC, it has primarily focused on able-bodied individuals. Notable exceptions include the studies by Krasoulis et al. (2017, 2020) [5,21], which explicitly investigated classification-based prosthesis control with Amputee participants. However, these studies included only two subjects and focused primarily on real-time control feasibility using optimally selected sensor subsets. Our study extends their work by analyzing a larger sample of participants with amputation. The ANOVA results in the current study confirmed a significant main effect of participant group, indicating that overall classification performance differed significantly between amputated and able-bodied participants (*p* < 0.001). Specifically, able-bodied participants achieved higher mean accuracies across modalities, suggesting that differences in physiological and mechanical conditions strongly influenced signal reliability and model performance. In amputee participants, reduced muscle mass, altered motor-unit recruitment, and spatially shifted EMG activity due to amputation can degrade signal separability. Additionally, residual limb mechanics introduce variability in ACC measurements, further impacting classifier stability. In contrast, able-bodied participants exhibit more consistent muscle activation patterns and limb kinematics, facilitating better training–testing alignment.

While differences between participant groups were statistically significant, their effect size was notably much lower than the effect size of sensor modality. Thus, the results support the use of ACC-only models for grasp type detection in both individuals with and without amputation. It also suggests that findings from able-bodied studies that use acceleration models may be useful to extend to individuals with amputations.

Nonetheless, by separately identifying the most predictive features for each group, we provide novel insights into how sensor contributions vary across user populations. Notably, the mean absolute value (MAV) of EMG consistently emerged as the most important feature when using EMG alone or in combination with ACC in both groups. This result aligns with findings from Tripathi et al. [16], who also reported MAV as well as the mean absolute envelope as key predictors for air writing gestures (i.e., letters written with fingers in free space). The root mean square (RMS) of both EMG and ACC also proved informative, which is also consistent with prior work on sign language recognition [11]. As average-based metrics, MAV and RMS are effective in capturing overall muscle effort within a time window, demonstrating their value as robust and interpretable features for gesture classification.

When using only ACC data, the top features differed slightly by population: the minimum (Min) was the most important for able-bodied participants while the mean was the most important for Amputees. Both features had importance scores exceeding 20%. The RMS, mean, MAV, min, and max were all significant ACC features for amputee and able-bodied users, which is consistent with prior able-bodied participant literature [11]. For participants with amputation, feature selection using both EMG and ACC data yielded only three top features, with a substantial increase in accuracy after adding ACC mean and max to EMG MAV (Figure 4c). This result indicates that a small but carefully chosen feature set can yield high performance, especially when including ACC-derived features. It suggests that effective prosthetic control for individuals with amputation may be achieved with minimal sensing and computational resources, which is an important step toward making such systems more practical, low cost, and user-friendly.

Well-chosen features are key to maximizing classifier performance and efficiency for both participants with and without amputation. The performance of our EMG+ACC classifier was comparable across participant groups, achieving maximum accuracy of 91.9% for able-bodied and 91.0% for Amputees using selected features, and 94.3% and 93.9%, respectively, when using all features. This performance is in line with results from Atzori et al. (2014) [7], who reported similar accuracies (maximum performance of 92.1% for able-bodied and 88.9% for Amputee participants) using just one EMG feature (marginal Discrete Wavelet Transform-mDWT) and one ACC feature (mean) for 40 able-bodied and five Amputee participants. ACC mean was also the most important feature for our Amputee group.

Notably, when identifying the most important features for the EMG+ACC condition, able-bodied participants relied heavily on EMG: the top seven features were EMG-derived, with ACC features (mean and minimum) added last. Despite their late inclusion, these ACC features still improved model accuracy by 20% and 10%, respectively—suggesting a complementary role even in able-bodied users. Without these ACC features, the model’s performance was comparable to the EMG-only model. It should also be noted that the selected feature subset for the able-bodied group was three times larger than the subset for the population of participants with amputation. This finding suggests that a wider range of EMG features contributed meaningfully to the classification accuracy in able-bodied individuals, likely due to more reliable and distinct muscle activation patterns. Still, a notable improvement in accuracy does not occur until adding at least two ACC features. This difference may reflect the greater reliability and discriminative power of EMG signals in able-bodied individuals, while also highlighting the utility of ACC features for both groups. In fact, adding those two ACC features to the able-bodied LDA model after the seven EMG features improved the accuracy from about 55% to nearly 90%.

In addition, ACC clearly serves as a more robust alternative in individuals with amputation, where EMG may not fully capture muscle activity due to the reduced quality or consistency of EMG signals from the residual limb. For participants with amputation, two of the top three selected features in the EMG+ACC condition were derived from ACC. This result indicates that ACC data may provide more robust or complementary information when EMG signals are compromised due to muscle atrophy, inconsistent electrode placement, or re-innervation patterns. This is also supported by previous work showing that the inclusion of inertial sensing improves real-time control performance [5]. These findings have immediate clinical implications. ACC sensors’ lower cost, simpler integration, and reduced sensitivity to daily placement variations make them ideal for prosthetic applications where reliability and user-friendliness are crucial.

While our findings show that ACC alone achieved performance comparable to the combined EMG+ACC condition, this outcome should be interpreted as task-specific rather than universally optimal. The results suggest that ACC can provide sufficient information for grasp-type recognition when EMG signals are unstable or impractical to acquire, such as in cases of electrode displacement, skin impedance changes, or perspiration. Nevertheless, combining EMG and ACC remains advantageous when feasible, as multimodal sensing typically enhances robustness and adaptability in real-world prosthetic control.

Finally, there is always a trade-off between average classification accuracy and training time. Using the selected feature subset cuts model training time by at least half of the training time with all features (Table 3). In some cases, such as in EMG+ACC for participants with amputation, the training time with selected features (2.45 s) was a sixth of the the time with all features (14.5 s). Adding more features increases training time but typically improves accuracy. This increase plateaus after the most important features are added (Figure 4). For each group of participants, the minimal feature was selected based on the trade off between improving accuracy at the cost of additional training time. Although the time savings may appear relatively minor on a high-performance desktop, such as the one used in this study, reducing training time by several seconds can be critical for real-time systems, iterative training, or operation on lower-power devices.

### Limitations

This work was not without limitations that should be acknowledged to inform future studies. First, the utilized dataset had been collected with sensor placement that did not target specific muscles, but used equidistant array placement. Investigating and optimizing sensor placement can potentially enhance signal accuracy, improve classification performance, and help designing efficient prosthetic devices with a minimal number of required sensors.

Second, the LDA classifier in this paper was trained on a participant-specific basis. There is a trade-off between participant-specific models, which tend to have better performance, and general models trained across different individuals, which may be more generalizable and don’t require re-training. Amputees may have unique differences based on the location and type of amputation, requiring subject-specific models. The utilized dataset included individuals with trans-radial amputations; the findings may not generalize well to other amputation levels (e.g., trans-humeral) or limb configurations, where muscle availability and limb kinematics may differ substantially.

Third, this work focused on only EMG and ACC data, as they are the most fundamental and widely adopted modalities for muscle activity and motion data [1,7,11,17,27]. It is possible that other sensing modalities, like force myography (FMG) [31,32], optical myography (OMG) [33,34], or hybrid FMG/OMG–IMU systems [35], may have better performance.

Next, this study utilized only one dataset, and findings may be specific to the tested population. Nonetheless, the selected MeganePro dataset is the largest open-access resource providing synchronized EMG and ACC recordings from both able-bodied and amputee participants. Other relevant datasets include different subsets of NinaPro dataset [36]. The generalizability of our findings could be further verified as more data is collected in future studies.

Finally, the gesture recognition performed in this study only considered grasp type detection for both static and dynamic task, involving object manipulation, but it did not consider object detection nor grip force detection. This is important for real-world prosthetic applications when intended grasp type and grip force are both needed for successful task completion (e.g., holding a grape vs a pencil). The inclusion of object and grip-force intent detection may increase the importance of EMG signals relative to ACC. The dataset utilized did not include force data. Incorporating both EMG and force measurements can enable more intuitive and functionally effective prosthetic control strategies, particularly in tasks requiring fine manipulation and variable force application.

## 5. Conclusions

In this study, we explored the efficacy of two different types of wearable sensors, sEMG and accelerometers, in predicting grasp types to enhance the functionality of active prosthetic devices. Evaluating EMG and ACC data in isolation enabled a clear comparison of their relative contributions to grasp-type prediction and established a baseline for future multimodal integration. Accelerometer signals outperformed EMG signals, with a greater improvement observed in participants with amputation than in able-bodied individuals. This improved performance supports the inclusion of accelerometers as the preferred sensor modality for classifying grasp types in prosthetic applications. Feature importance analysis further revealed that a small, carefully selected subset of features, especially from ACC, was sufficient to maintain high classification accuracy while reducing training time. Among all features, MAV, WL, AAC, VAR, and RMS consistently ranked as the most informative for EMG data, while mean, Min, Max, and RMS were most important for ACC. Participants with amputation, in particular, required only three features in the EMG+ACC condition to achieve near-maximal performance, underscoring the value of a minimal yet effective feature set. The findings suggest that minimizing the types of sensors, as well as reducing the number of input features, are feasible to avoid unnecessary cost and complexity in design and usage of efficient active hand prosthetic devices. Future work should investigate generalizing classification models across different participants, perhaps using transfer learning techniques, and reducing the number of required sensor electrodes through channel selection. An investigation of the combined object and/or grasp force recognition with grasp classification is needed for truly applicable intent detection. Exploring other sensing modalities and their combinations could reveal even more optimal sensing suites. Finally, the models should be tested in real-time prosthesis applications with users with upper-limb amputation. Ultimately, this work highlights the importance of ACC data in the goal of creating more intuitive and efficient multi-degree-of-freedom prosthetic devices that seamlessly integrate user intent for an improved user experience.

## Figures and Tables

**Figure 1 sensors-25-06984-f001:**
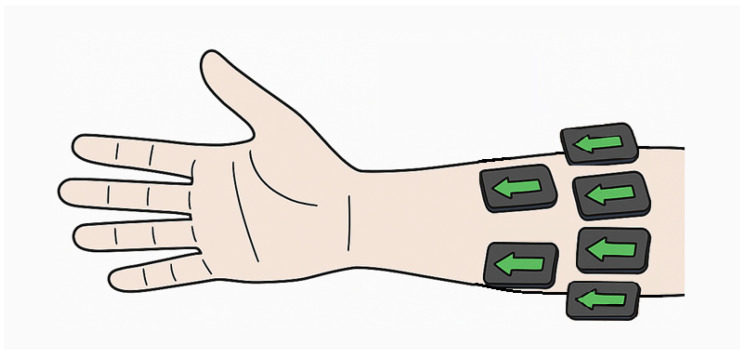
Schematic of forearm sensor placement during data collection. Arrows indicate x-axis direction, parallel to the forearm.

**Figure 2 sensors-25-06984-f002:**
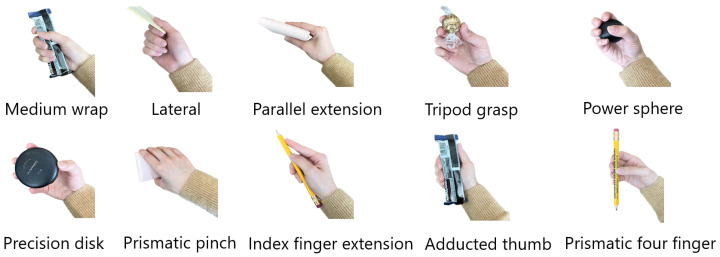
Ten different grasp types used as the classifier labels.

**Figure 3 sensors-25-06984-f003:**
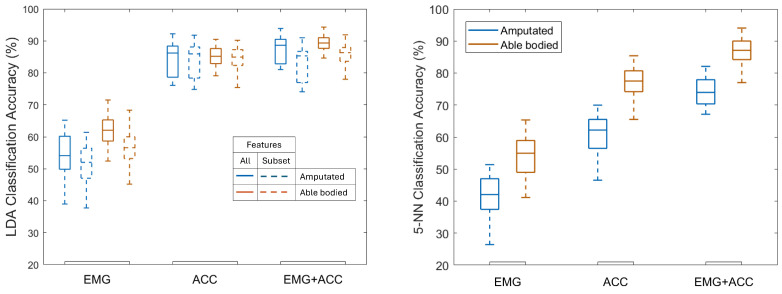
Test set classification accuracy for Linear Discriminant Analysis (LDA, **left**) using all features and feature subsets (Table 2), and 5-nearest neighbors (5-NN) classifier (**right**) trained using all features, across EMG, ACC, and EMG+ACC modalities for both participant groups (amputated and able-bodied). LDA outperformed 5-NN, but both followed a similar overall trend across sensing modalities.

**Figure 4 sensors-25-06984-f004:**
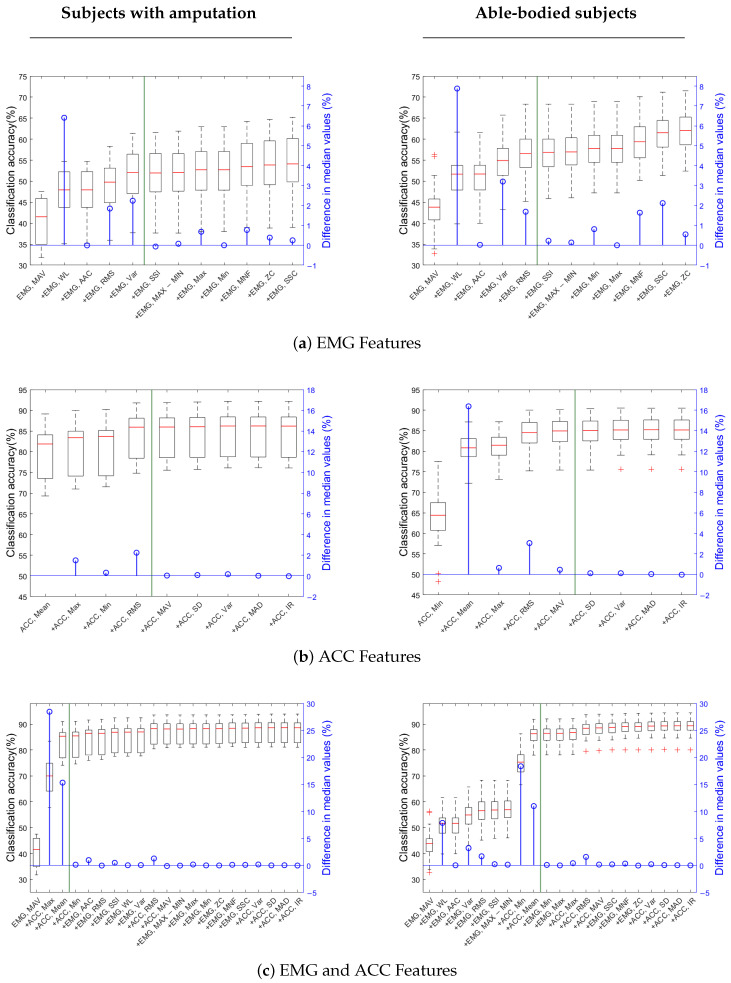
Accuracy curve with adding feature groups across subjects for (**a**) EMG only data, (**b**) ACC only data, (**c**) both EMG and ACC data. Three plots on the left column are for subjects with amputation and three plots on the right column are for able-bodied subjects. Each boxplot reports the median, the first and third quartiles, and the extrema. The green vertical line in each plot indicates the stopping point for feature selection.

**Figure 5 sensors-25-06984-f005:**
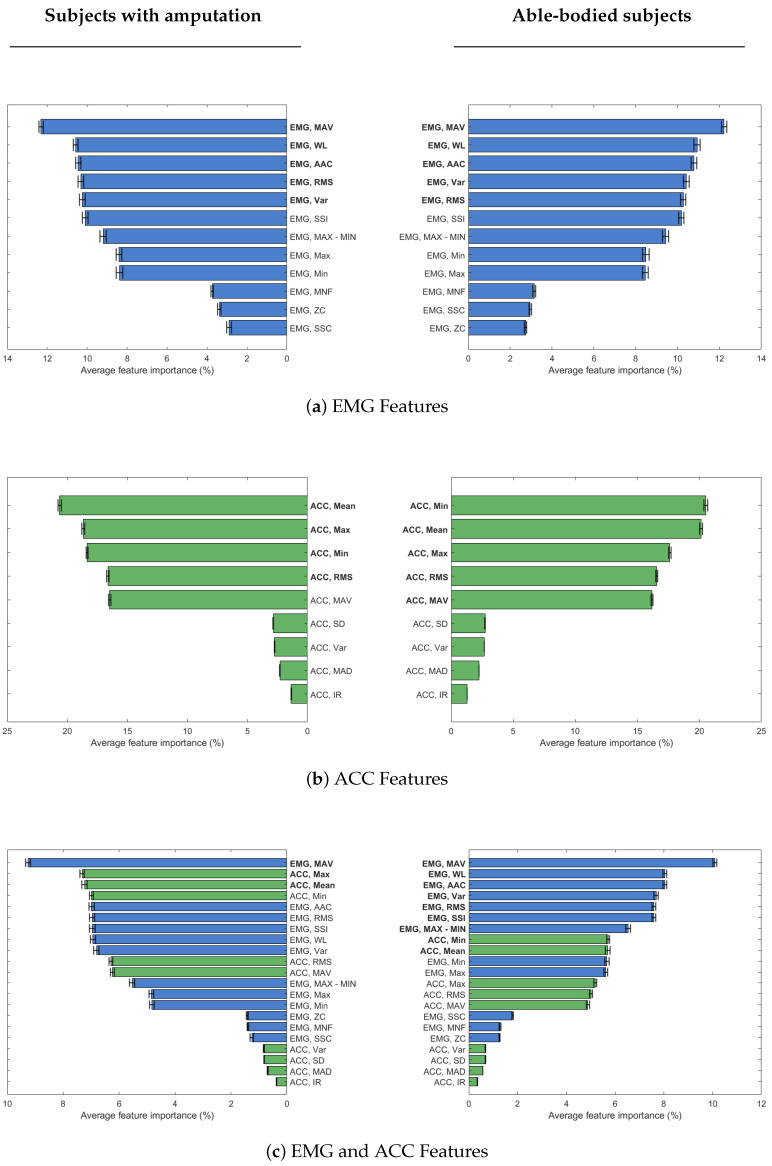
Sorted mean feature importance percentage across channels for subjects with and without amputation when evaluating features for (**a**) EMG only data, (**b**) ACC only data, and (**c**) both EMG (blue) and ACC (green) data. Three plots on the left column are for subjects with amputation and three plots on the right column are for able-bodied subjects. The features in **bold** font are the selected set of features for each modality and participant group.

**Table 1 sensors-25-06984-t001:** All features calculated for EMG and ACC data.

Feature	Acronym	Sensor	References
Zero crossings	ZC	EMG	[10,13]
Slope sign changes	SSC	EMG	[10]
Waveform length	WL	EMG	[7,10,11,12,13]
Max-min	MAX-MIN	EMG	[13]
Average amplitude change	AAC	EMG	[11,13]
Simple square integral	SSI	EMG	[11,13]
Mean frequency	MNF	EMG	[11,13]
Mean	Mean	ACC	[7,11,12,17]
Standard deviation	SD	ACC	[11,12,17]
Mean absolute deviation	MAD	ACC	[12,17]
Interquartile range	IR	ACC	[12]
Mean absolute value	MAV	EMG, ACC	[10,11,12,13]
Variance	Var	EMG, ACC	[11,12,13]
Maximum value	Max	EMG, ACC	[12,13]
Minimum value	Min	EMG, ACC	[12,13]
Root mean square	RMS	EMG, ACC	[7,10,11,13,17]

**Table 2 sensors-25-06984-t002:** Selected features for each modality.

Subjects with Amputation		
EMG	ACC	EMG+ACC
MAV	Mean	EMG, MAV
WL	Max	ACC, Max
AAC	Min	ACC, Mean
RMS	RMS	–
Var	–	–
Able-bodied subjects		
EMG	ACC	EMG+ACC
MAV	Min	EMG, MAV
WL	Mean	EMG, WL
AAC	Max	EMG, AAC
Var	RMS	EMG, Var
RMS	MAV	EMG, RMS
–	–	EMG, SSI
–	–	EMG, MAX-MIN
–	–	ACC, Min
–	–	ACC, Mean

**Table 3 sensors-25-06984-t003:** Comparison of the LDA classifier training times in seconds (s) between all features and selected input features for different sensor modalities and participant groups.

Training Time for Participants with Amputation (s)		
Modality	All Features	Selected features
EMG	3.77	1.77
ACC	11.83	5.25
EMG+ACC	14.50	2.45
Training time for able-bodied participants (s)		
Modality	All features	Selected features
EMG	4.38	1.74
ACC	10.52	5.11
EMG+ACC	17.91	5.16

## Data Availability

The data presented in this study were derived from the publicly available MeganePro dataset [10]. This dataset is available in the Harvard Dataverse repository at https://doi.org/10.7910/DVN/1Z3IOM, reference number [37].

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
