# Peer review of "Simplifying Prediction of Intended Grasp Type: Accelerometry Performs Comparably to Combined EMG-Accelerometry in Individuals With and Without Amputation"

_sensors, 2025, doi:10.3390/s25226984_

Round 1

Reviewer 1 Report

Comments and Suggestions for Authors

The approach is goal-oriented and covers an interesting question.

The data set reference is from 2020. It contains a.o. EMG and accelerometer data. The two research questions addressed are: (i) which are the most important ones for grasp type recognition and (ii) which features contribute most to accuracy for the two populations of (i) amputees and (ii) able-bodied people.

The discussion argues that the current study fully addresses the two research questions while other studies are only doing this partially. That is the novelty.

  • Methodologically there is nothing new.
  • The feature table does not contain any unexpected features, So also here: no novelty.

The paper is well-structured and studies the 2 research questions in detail.

  1. From the start on, subject specific LDA classifiers are selected. There is no motivation why only LDA
  2. Accuracy might be important, but there is no indication of misclassification (confusion matrices for example)
  3. Several comparisons are made in the discussion with other references, but it is unclear if these have used a similar measurement setup as the one of the used data set.
  4. The training and testing procedures are well-described, but they all occur on data from the same publicly available data set. An independent validation on another data set would improve the credibility and reveal the real robustness en generalization capacity of the results of the findings.
  5. The methodology contains some unclear steps for me that could be better explained, I presume:
  • Line 159 “It is wort nothing…..repetitions” what is exactly meant by that?
  • The use of the accuracy median difference to define a stop criterium for feature selection? How is it put in practice (numerical stop criterion?)  and how does it explain the green vertical lines in Figure 4?

PUNCTUAL remarks:

Line 188: without amputation -- > with amputation

Table 3: add units in the caption of the Table

Reviewer 2 Report

Comments and Suggestions for Authors

Strong Points

The study's use of data from individuals with amputation is commendable and socially meaningful, as it directly contributes to improving assistive technologies.
The attempt to compare electromyography (EMG) and accelerometer (ACC) signals for grasp-type prediction is a relevant direction in the field of wearable sensing for prosthetic control.

Weak Points

1. Limited Scope of Sensor Comparison
The study focuses solely on comparing EMG and ACC, without discussing other wearable biosensing modalities capable of capturing muscle activity. A more comprehensive discussion—including FMG (Force Myography), OMG (Optical Myography), and/or FMG/OMG combined with IMU is necessary.
Without this broader context, it is difficult to interpret the conclusion that one modality is "more informative" for grasp-type prediction.

2. Questionable Practical Conclusion
The claim that using only ACC is sufficient is debatable. In practice, combining EMG and ACC sensors poses no major hardware limitations, and prior studies have shown that sensor fusion improves robustness and accuracy.
If the authors intentionally avoided EMG due to practical issues (e.g., electrode attachment difficulty), this should be explicitly stated. Or/and, they should discuss other muscle-activity sensing technologies that could mitigate such limitations. For example, the combination of Infrared Optical Sensors and an IMU are practical as an arm-band type sensor system (OMG/FMG with IMU is still relatively low cost).

3. Influence of Amputation Characteristics
The degree and location of amputation likely influence the predictive performance of the algorithm, but this aspect is not analyzed. Clarifying how the amputation level affects signal quality and classification results would add depth to the study.
The dataset appears to have a small number of amputee participants—could more samples be included, or could intra-dataset variability be analyzed for insight into individual differences?

4. Lack of Statistical Validation
The study lacks statistical analyses, such as significance testing or effect size estimation. These are essential to confirm whether observed differences between modalities are meaningful rather than incidental.

5. Limited Model
The experiments use a relatively simple model, and the evaluation seems insufficient to support general conclusions. In addition, there might be a good combination between an algorithm and bio-signal types.

6. Evaluation Metrics
The paper only reports accuracy, but omits key metrics such as precision, recall, and F1-score, which are necessary to comprehensively evaluate classification performance.

7. Insufficient Discussion on Group Differences
The difference between amputee and non-amputee participants deserves deeper discussion. Understanding why and how the performance varies between groups could yield valuable insights into sensor reliability and algorithm adaptability.

Round 2

Reviewer 1 Report

Comments and Suggestions for Authors

I think the overall quality of the paper has been improved. The results are now more credible, so the "interest to readers" has improved. The "significance of the content" and hence the "overall merit" for the application considered in the paper have become average. The methodological novelty/originality is still low,

Seen the appreciations mentioned above, I do not want to reject it a second time

Reviewer 2 Report

Comments and Suggestions for Authors

The authors have appropriately expanded the discussion to include other sensing modalities such as FMG and OMG. However, this section currently lacks supporting references. It would strengthen the careful and thoughtful discussion to cite representative works on FMG, OMG, or hybrid sensing systems (e.g., FMG/IMU, OMG/IMU) to provide proper context for these technologies.

The authors have added statistical validation using a two-way mixed-design ANOVA, which is appreciated. However, the statistical section does not report effect sizes (e.g., partial η² or Cohen’s d). Including effect size measures would provide a clearer understanding of the practical significance of the observed differences across modalities and participant groups.
